# Anti-Dyslipidemic and Anti-Diabetic Properties of Corosolic Acid: A Narrative Review

**Rossella Cannarella** [1,2,*] , **Vincenzo Garofalo** [1] and **Aldo E. Calogero** [1,*]

1 Department of Clinical and Experimental Medicine, University of Catania, 95123 Catania, Italy; vgarofalo985@gmail.com
2 Glickman Urological & Kidney Institute, Cleveland Clinic Foundation, Cleveland, OH 44195, USA
* Correspondence: rossella.cannarella@phd.unict.it (R.C.); aldo.calogero@unict.it (A.E.C.)

**Abstract:** Corosolic acid (CA), a natural compound derived from the Banaba tree (*Lagerstroemia speciosa*), has attracted attention for its potential therapeutic properties in the management of metabolic diseases. This narrative review aims to summarize the current evidence on the anti-dyslipidemic, anti-diabetic, and anti-inflammatory effects of CA and to understand the pharmacokinetics and molecular mechanisms through the analysis of preclinical and clinical studies.

**Keywords:** corosolic acid; dyslipidemia; diabetes

## 1. Introduction

Metabolic disorders are a heterogeneous group of conditions characterized by metabolic dysregulation, which include a wide range of diseases, such as dyslipidemia, liver steatosis, metabolic syndrome (MetS), insulin resistance, prediabetes, and diabetes mellitus (DM). These conditions often coexist and share common underlying mechanisms. Their prevalence in the general population is remarkable.

World Health Organization (WHO) data show that the global prevalence of dyslipidemia in adults aged ≥25 years in 2008 was ~39%. Approximately one-third of deaths from heart attack or ischemic stroke were attributed to elevated plasma LDL-cholesterol levels, which are higher in men than in women [1,2]. The prevalence of dyslipidemia can change depending on various factors such as age, gender, ethnicity, and geographical location. Similarly, the global prevalence of non-alcoholic fatty liver disease (NAFLD) was estimated to be about 25%, with a progression toward about 40% [3], while the global prevalence of MetS is impressively high, being estimated to be about a quarter of the world population. However, the prevalence estimate varies depending on the criteria used for its diagnosis, such as International Diabetes Federation (IDF) or National Cholesterol Education Program (NCEP) Adult Treatment Panel (ATP) III [4].

As for prediabetes, its prevalence depends on the age of the population examined and the diagnostic criteria used, such as American Diabetes Association or WHO. It is estimated that 86.1 million adults in the United States have prediabetes [5]. Lastly, as reported by the WHO, in recent decades, there has been an exponential growth in the prevalence of DM of all types worldwide. The prevalence of DM, indeed, increased from 108 million (4.7%) in 1980 to 425 million (8.5%) in 2017, and it is estimated to be 629 million by 2045 [6].

Metabolic disorders are associated with cardiovascular diseases and mortality. Accordingly, dyslipidemia is a significant risk factor for atherosclerotic cardiovascular disease (ASCVD), such as heart disease and stroke [7]. Furthermore, MetS is often a precursor to the development of DM and cardiovascular complications [8].

The high prevalence of metabolic diseases worldwide, as well as their predictive role in the onset of cardiovascular disorders, justifies the importance of a proper therapeutic approach. Treatments include lifestyle modifications and pharmacological intervention. The latter includes statins, fibrates, insulin, metformin, dipeptidyl peptidase-4 inhibitors

(DPP-4is), and, more recently, glucagon-like peptide-1 receptor agonists (GLP-1 RAs) and sodium–glucose co-transporter 2 inhibitors (SGLT-2is). Although important for metabolic control and cardiovascular and kidney health, these drugs can cause adverse events, such as hypoglycemia, urogenital infections, hypotension, and others [9].

In recent years, the beneficial role of nutraceutical compounds in metabolic diseases has emerged. Nutraceuticals are foods or their components that have the ability to exert beneficial effects on health, contributing to the prevention and/or treatment of diseases [10]. Several of these natural compounds have various benefits on metabolism including cholesterol-lowering, insulin-sensitizing, anti-hypertensive, cardioprotective, weight-lowering, anti-oxidant, and anti-inflammatory effects. The use of nutraceuticals, combined with a healthy lifestyle and pharmaceutical interventions, has shown promising effects in supporting overall health and potentially complementing traditional therapies [11]. The main nutraceutical compounds available and approved for clinical practice are reported in Table 1.

**Table 1.** Main nutraceutical compounds used in clinical practice.

| Nutraceutical Compounds | Effects | References |
|---|---|---|
| Flavonoids | ↑ NO availability; ↓ ROS formation; ET-1 inhibition; ↓ ACE activity; anti-inflammatory activity; ↑ insulin sensitivity | Serafini et al., 2010 [12] |
| Poly-unsaturated Fatty Acids (PUFAs) | Anti-inflammatory activity; ↑ PG vasodilators; ↑ NO synthase; ↓ insulin resistance | Simopoulos, 2002 [13] |
| L-Arginine | ↑ NO availability; anti-oxidant action | Silva et al., 2017 [14] |
| Lycopene | Anti-oxidant action; free radical scavenger | Leh and Lee, 2022 [15] |
| Resveratrol | Anti-oxidant action; anti-inflammatory activity; ↑ NO availability | Zhou et al., 2021 [16] |
| Allicin | ↑ NO availability; ↓ ACE activity; ↑ insulin sensitivity; ↑ HDL-C | Borlinghaus et al., 2014 [17] |
| Monacolines | HMG-CoA-reductase inhibition; ↓ LDL-C; ↑ HDL-C | Xiong et al., 2019 [18] |
| Inositols | ↓ LDL-C; ↑ insulin sensitivity | Bevilacqua and Bizzarri, 2018 [19] |
| Berberine | ↑ hepatic LDL-C receptors; ↑ insulin receptors | Song et al., 2020 [20] |
| Beta-glucans | ↓ intestinal absorption of cholesterol; ↑ insulin sensitivity | Ciecierska et al., 2019 [21] |
| Polyphenols | ↓ intestinal absorption of glucose; protective action on β cells; ↓ hepatic glucose production; direct action on GLUT4 | Khan and Mukhtar, 2018 [22] |
| Phytosterols | ↓ intestinal absorption of cholesterol; ↓ LDL-C | Nattagh-Eshtivani et al., 2022 [23] |
| Silymarin | ↑ insulin sensitivity; ↓ hepatic inflammation; ↑ liver protein synthesis | Vahabzadeh et al., 2018 [24] |
| Vitamin E | Anti-oxidant properties; ↑ cell renewal | Miyazawa et al., 2019 [25] |
| Astaxanthin | ↓ lipogenesis; ↓ insulin resistance; ↓ hepatic inflammation | Chang and Xiong, 2020 [26] |
| Curcumin | ↑ insulin sensitivity; anti-oxidant action; anti-inflammatory activity | Peng et al., 2021 [27] |
| Betaine | Hepatoprotection; ↓ homocysteinemia | Chen et al., 2022 [28] |
| Cinnamaldehyde | Hypoglycemic, cholesterol-lowering, and anti-hypertensive activity | Zhu et al., 2017 [29] |
| Corosolic Acid | ↑ insulin sensitivity; ↓ body weight; ↓ LDL-C | Zhao et al., 2021 [30] |

Abbreviations. ACE: Angiotensin-Converting Enzyme; ET-1: Endothelin-1; HDL-C: High-Density Lipoprotein Cholesterol; LDL-C: Low-Density Lipoprotein Cholesterol; HMG-CoA: Hydroxymethylglutaryl-CoA; NO: Nitric Oxide; PG: Prostaglandin; ROS: Reactive Oxygen Species. ↓, decrease; ↑, increase.

The use of nutraceuticals in clinical practice has several benefits. They are typically derived from natural sources such as plants, herbs, and food components, often considered safe and well-tolerated when used appropriately, especially when compared to some pharmaceutical drugs that may have adverse effects or interactions. On the other hand, a significant concern for nutraceuticals is the lack of strict regulation and quality control measures compared to drugs. The dietary supplement industry operates under different regulations that do not require premarket approval or rigorous testing for safety and efficacy. This lack of oversight raises concerns about inconsistent product quality, inaccurate labeling, and the presence of impurities or contaminants [31]. Another disadvantage of nutraceutical compounds is the limited scientific evidence supporting their purported health benefits because many of the studies have been conducted on small sample sizes. The lack of well-designed, large-scale clinical studies makes it difficult to assess their true efficacy, optimal dosages, and potential interactions with other drugs [31].

Corosolic acid (CA), also known as $2\alpha$-hydroxy-ursolic acid, is an ursane-type penta-cyclic triterpenoid found naturally in several plant species, including *Lagerstroemia speciosa* (Banaba) [32]. It has attracted considerable attention for its potential health benefits and therapeutic properties, such as anti-diabetic, anti-obesity, anti-hyperlipidemic, anti-inflammatory, and anti-tumor effects [30]. CA is characterized by a complex structure consisting of five fused rings. The chemical formula of CA is $C_{30}H_{48}O_4$. Its structural analogs, ursolic acid, oleanolic acid, maslinic acid, asiatic acid, and betulinic acid, exhibit activities like those of CA [33]. CA has shown promising anti-diabetic properties. It has been found to enhance glucose uptake, increase insulin sensitivity, and inhibit enzymes involved in carbohydrate absorption, such as $\alpha$-amylase and $\alpha$-glucosidase. These mechanisms contribute to improved glycemic control and have potential implications in the management of DM and its associated complications [34]. Furthermore, CA inhibits adipogenesis and promotes lipolysis, contributing to a reduction in body weight gain and adipose tissue accumulation. Moreover, CA appears to regulate lipid metabolism by reducing triglyceride levels and increasing high-density lipoprotein (HDL) cholesterol, thus suggesting a potential role in managing obesity-related lipid abnormalities [35]. Finally, CA possesses anti-oxidant and anti-inflammatory properties, which are associated with its potential protective effects against metabolic disorders. It has been shown to reduce oxidative stress, suppress pro-inflammatory cytokines, and inhibit inflammatory signaling pathways. These effects contribute to the prevention or attenuation of metabolic disorders and associated complications, including insulin resistance and chronic inflammation [30], and may support the use of CA in these diseases.

Since no comprehensive review has been published thus far attempting to evaluate the anti-diabetic and anti-dyslipidemic properties of CA, we provide here a comprehensive report of the molecular mechanisms by which CA is able to influence glucose and lipid metabolism. To achieve this, we reviewed evidence acquired from in vivo studies in both animal models and humans.

## 2. Molecular Mechanisms

### 2.1. Pharmacokinetics

CA is a naturally occurring triterpenoid compound found in several medicinal plants, including Lagerstroemia speciosa. CA has gained attention for its potential pharmacological effects, particularly its anti-diabetic, anti-obesity, anti-inflammatory, and anti-cancer activities. Understanding the pharmacokinetics of CA is essential to optimize its therapeutic use and evaluate its safety profile. After oral administration, about 30 percent of CA is absorbed by the gastric mucosa, because being partly a weakly acidic compound, it occurs in free form within the acidic environment of the stomach. An experimental paradigm using an exposed rat stomach, cut at the pylorus, washed with saline, and then treated with 2 mL of gastric perfusion solution containing increasing concentrations of CA showed that absorption at concentrations of 5 mg/mL and 10 mg/mL was higher than when the administered concentration of CA was 20 mg/mL. This suggests that absorption at the

level of gastric cells does not occur only by passive transport but mechanisms of active absorption may also be involved [36].

In the intestine, by cannulation of different intestinal segments and perfusion with a CA-containing solution, the absorption constant (Ka) was found to be similar in different segments with good effective permeability (Peff), particularly in the jejunum and ileum compared with the duodenum and the colon. It has been observed that Peff is directly proportional to the concentration of CA and as one increases the other also increases. This suggests that active transport mechanisms are also involved in intestinal absorption [18].

After absorption, the plasma concentration of CA is only 669 ng/mL at doses of 10 mg/mL, thus indicating low bioavailability [37]. Furthermore, the concentration is higher in the plasma of the hepatic portal circulation than in the jugular plasma [36]. Five possible metabolites of CA have been identified in both plasma and bile, suggesting hepatic metabolism. These included products of methyl-carboxylation, methyl-hydroxylation at C24 or C25, and glucuronidation (M1, M2, and M3). A metabolite not found in bile (M4), characterized by a methyl-aldehyde substitution at C24 or C25, was also identified in plasma. Furthermore, a metabolite produced by glucuronidation and acetylation of CA was found only in bile (M5). Thus, it can be assumed that CA is metabolized via methyl-carboxylation, hydroxylation, methyl-aldehyde substitution, glucuronidation, and acetylation at the level of liver cells [36].

Drug metabolism includes phase I (oxidation, reduction, and hydrolysis) and phase II (glucuronidation, sulfation, acetylation, and methylation) [38]. The main enzymes that catalyze phase I reactions in liver microsomes belong to the cytochrome P450 family and, among these, CYP1A2 and CYP3A4 are mainly responsible for the oxidation of CA and the formation of M1, M2, and M4 in plasma. Phase II enzymes such as glucuronyltransferase and acetyltransferase are instead involved in the formation of M3 in both plasma and bile and M5 only in bile [36,39]. The presence of these metabolites could contribute to the various functions of CA. Zhang and colleagues showed that plasma containing these metabolites stimulates glucose consumption and significantly inhibits the growth of tumor cells, demonstrating their possible anti-diabetic and anti-tumor effects [36]. However, further studies are needed to understand their mechanism of action.

*2.2. Biological Properties*

CA attracts interest for its various biological properties and potential therapeutic applications. One of the most notable effects of CA is its anti-diabetic activity. Several studies have demonstrated its ability to improve glucose uptake and utilization, making it a good candidate to help in the management of diabetes and related metabolic disorders [40]. Furthermore, CA exhibits potent anti-oxidant activity. It has been shown to scavenge free radicals, reduce oxidative damage, and protect cells from damage induced by oxidative stress [30]. Furthermore, studies have revealed that CA inhibits the production of pro-inflammatory mediators, including cytokines and chemokines, and exerts anti-proliferative effects by inducing cell cycle arrest and apoptosis, inhibiting angiogenesis, and suppressing metastasis [30].

## 3. Effect of Corosolic Acid: Preclinical Evidence

*3.1. Anti-Inflammatory and Anti-Oxidant Effects*

CA has shown promising effects in the modulation of inflammation and oxidative stress. These properties make it a potentially valuable therapeutic agent for the management of inflammatory conditions and diseases related to oxidative stress. CA exerts its anti-inflammatory activity by inhibiting the production of pro-inflammatory mediators, such as cytokines and chemokines, and by suppressing the activation of inhibitor of IL-1 receptor-associated kinase 1 (IRAK1) and nuclear factor-kappa B (NF-κB), resulting in reduced expression of monocyte chemoattractant protein-1 (MCP-1), NLR family pyrin domain-containing 3 (NLRP3), and interleukin-1 (IL-1), and therefore the inflammatory response [40]. The anti-inflammatory effects of CA also lead to an improvement in insulin

resistance. Through the activation of adenosine monophosphate-activated protein kinase (AMPK) in a hepatic kinase B1-dependent manner, CA reduces the inhibitor of nuclear factor-kappa B kinase (IKKβ) and pro-inflammatory cytokines, inhibits macrophage infiltration and inflammation, and upregulates liver kinase B1 (LKB-1), insulin receptor substrate 1 (IRS-1), and protein kinase B (Akt), thus reducing the insulin resistance [30].

Kim and colleagues demonstrated that CA reduces acute inflammation by regulating IRAK1 phosphorylation via a nuclear factor-kappa B (NF-κB)-independent pathway [41]. Phosphorylated IRAK1 stimulates the release of inflammatory cytokines and controls inflammasome assembly and activation during acute inflammation [42]. In their study, Kim and colleagues showed that CA inhibits lipopolysaccharide-induced phosphorylation of IRAK1, thereby abolishing the formation of the inflammasome complex. These results suggest the possibility of using CA in acute inflammation, such as sepsis or endotoxin shock [41].

Li and colleagues demonstrated that the anti-inflammatory and anti-oxidant activity of CA is also manifested in the kidney, inhibiting the proliferation of glomerular mesangial cells (GMCs), with a protective effect against diabetic nephropathy in mice. Mechanisms involved in the proliferation of GMCs include the activation of PKC-mitogen-activated protein kinases (MAPKs) and the formation of reactive oxygen species (ROS) from nicotinamide adenine dinucleotide phosphate (NADPH) in reduced form (NOX). CA administration reduces MAPK activation and blocks the expression of NOX4, NOX2, and the NOX-associated subunits p22phox and p47phox by inhibiting the activation of ERK1/2 signal transduction, resulting in decreased ROS production and dose-dependent inhibition of GMC proliferation [43].

CA also plays a protective role in the liver by reducing ethanol-induced liver damage. CA inhibits ethanol-induced p38 and JNK MAPK signaling pathways and restores ethanol-suppressed autophagy through AMPK activation in a dose-dependent manner. This results in a reduced concentration of ROS in the cytoplasm of liver cells [44].

The anti-inflammatory and anti-oxidant effects of CA also occur at the level of endothelial cells in the arterial wall. Free radicals exert a direct action on endothelial cells and facilitate the adhesion of T cells and monocytes during the early stages of the atherogenic process. CA that reduces oxidative stress could be a useful tool in atherosclerosis-related diseases such as blood hypertension [45].

Metabolic conditions such as diabetes, dyslipidemia, oxidative stress, and inflammation increase the risk of cardiovascular disease. A recent study in a rat model with myocardial infarction demonstrated the cardioprotective effect of CA via the PPAR-γ pathway, which is involved in the regulation of glucose, fatty acid, and lipoprotein metabolism and inhibits the secretion of pro-inflammatory cytokines. CA administration would seem to increase PPAR-γ activity in myocardial cells, exerting anti-oxidant effects by decreasing ROS production, normalizing CK-MB and LDH levels, and stabilizing the cardiomyocyte cell membrane. These results highlight the role of PPAR-γ pathway activation in the cardioprotective effects of CA [46] (Figure 1).

### 3.2. Anti-Diabetic Activity

The potential anti-diabetic activity of CA is exerted in several ways. Takagi and collaborators showed that 30 min after oral administration, CA acts directly in the small intestine by inhibiting the hydrolysis of sucrose into glucose, without any effect on maltose and lactose [47]. A key enzyme involved in the hydroxylation of carbohydrates to produce glucose is α-glucosidase. Ni et al. have shown that CA is able to inhibit α-glucosidase activity in a non-competitive manner. Indeed, CA binds the entrance part of the active center of the enzyme through the interaction with the amino acids Ser157, Arg442, Phe303, Arg315, Tyr158, and Gln353, causing a conformational change of α-glucosidase and the failure to release the product of the catalytic reaction, thus suppressing the catalytic activity [48]. Another study evaluated the synergistic effects of CA and acarbose in inhibiting α-amylase and α-glucosidase [49].

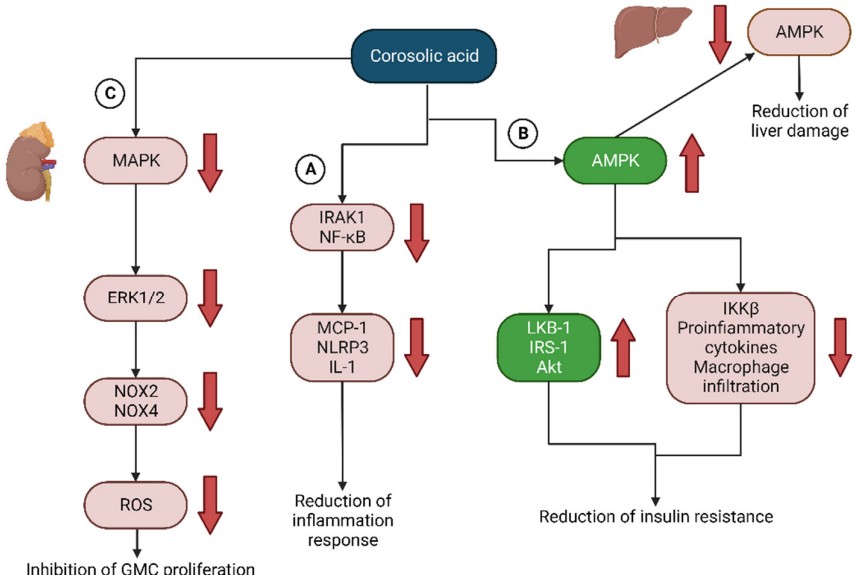

**Figure 1.** Anti-inflammatory and anti-oxidant activity. (**A**). CA suppresses the activation of inhibitor of IL-1 receptor-associated kinase (IRAK1) and nuclear factor-kappa B (NF-κB), resulting in reduced expression of monocyte chemoattractant protein-1 (MCP-1), NLR family pyrin domain-containing 3 (NLRP3), and interleukin-1 (IL-1), and thus the inflammatory response. (**B**). CA activates adenosine monophosphate-activated protein kinase (AMPK) and reduces the inhibitor of nuclear factor-kappa B kinase (IKKβ) and inflammatory cytokines, inhibits macrophage infiltration and inflammation, and upregulates liver kinase B1 (LKB-1), insulin receptor substrate-1 (IRS-1), and protein kinase B (Akt), thereby reducing IR. In the liver, CA inhibits the p38 and JNK MAPK signaling pathway through the activation of AMPK. (**C**). In the kidney, CA reduces MAPK activation and consequently the activation of ERK1/2 signal transduction, blocking the expression of NOX4, NOX2, resulting in a reduced ROS production and inhibition of GMC proliferation.

Another mechanism by which CA can reduce glucose levels is by increasing its uptake into cells. A study in an animal model increased glucose uptake after CA administration, reducing blood glucose levels. This effect could be due to an increased sensitivity of the cells to insulin. To validate this hypothesis, Shi and colleagues used a phosphoinositide 3-kinase (PI3K) inhibitor, thereby blocking the intrinsic insulin pathway. CA induced an increase in glucose uptake which was abolished by the PI3K inhibitor. This result confirms the potential role of CA in increasing glucose uptake at the cellular level [50]. The mechanism by which CA increases glucose uptake is determined by an increase in the sensitivity of insulin to its receptor. Insulin binding to its receptor brings the two β subunits closer together, allowing them to autophosphorylate the tyrosine-rich domain. The activated receptor in turn adds phosphate groups onto specific substrates, which activate other phosphate groups, allowing signal propagation. IRS-1 is one of these substrates, which activates the PI3K enzyme that activates the Akt/mTOR pathway and triggers the fusion of vesicles containing the glucose transporter GLUT4 with the plasma membrane, increasing glucose reabsorption [51–53]. CA appears to act on the insulin signaling pathway by increasing the level of tyrosine phosphorylation of the insulin receptor β in the presence of a submaximal insulin concentration (10 nM). Furthermore, the level of insulin-stimulated Akt Ser473 phosphorylation increases after CA administration [50]. These results suggest that CA enhances the insulin signaling pathway through increased phosphorylation of some substrates.

CA acts through a synergistic, but not additive, mechanism on glucose uptake and substrate phosphorylation. Indeed, it acts as an insulin sensitizer, increasing the effect of low concentrations of insulin but not the effect of saturated concentrations of insulin. Several non-receptor protein tyrosine phosphatases (PTPs) are able to reduce insulin sensitivity

through dephosphorylation of the insulin receptor β subunit [54–56]. CA may inhibit the activity of some PTPs such as protein tyrosine phosphatase 1B (PTP1B), T-cell protein tyrosine phosphatase (TCPTP), Src homology phosphatase-1 (SHP1), and Src homology phosphatase-2 (SHP2), thereby increasing the phosphorylation of the β subunit of the insulin receptor [50].

A study in rats analyzed the mechanism of action of CA on hepatic gluconeogenesis, whose regulation depends on the levels of fructose-2,6-bisphosphate (F-2,6-BP). The latter, in turn, is regulated by a bifunctional enzyme with two domains: phosphofructokinase 2 (PFK-2), which synthesizes F-2,6-BP, and fructose bisphosphatase (FBPase), which degrades it. If blood glucose levels are low, glucagon increases the concentration of cAMP and therefore PKA. This in turn phosphorylates and inactivates the enzyme domain of PFK-2 and activates FBPase, resulting in a decrease in F2,6-BP concentration and an increase in the gluconeogenic pathway. Conversely, if blood glucose levels are high, insulin is secreted and activates phosphoprotein phosphatase (PP1), which dephosphorylates and thereby activates PFK-2 and inactivates FBPase, resulting in increased F2,6P concentration and decreased gluconeogenic pathway [57]. Yamada and colleagues demonstrated that CA increases F-2,6-BP production by decreasing cAMP levels and inhibiting PKA in rat liver hepatocytes. As mentioned before, this results in a lack of FBPase activation and increased PFK-2 activity, which then leads to increased F-2,6-BP synthesis. By acting in this way, CA could inhibit gluconeogenesis [58].

Xu and colleagues conducted both in vitro and in vivo studies to investigate the molecular mechanisms by which CA acts. Phosphoenolpyruvate carboxykinase (PEPCK) is an enzyme that catalyzes the first step in gluconeogenesis. In most models of diabetes, increased expression of the *PEPCK* gene is observed in the liver, and its overexpression contributes to increased hepatic glucose production [59]. In HepG2 cell models with overexpression of the *PEPCK* gene, CA administration promotes glucose consumption by inhibiting PEPCK upregulation [60]. In a zebrafish model of T2D, CA administration reduced glycogen degradation and increased glucose consumption by regulating some key enzymes. A reduction in the expression of PEPCK, glucose transporter 1 (GLUT1), GLUT2, GLUT3, lactate dehydrogenase A (LDHA), lactate dehydrogenase B (LDHB), and insulin α (INS α) and increased levels of GP, glycogen synthase (GYS1), glucose-6-phosphatase (G6Pase), phosphofructo-2-kinase/fructose-2,6-bisphosphatase 3 (PFKFB3), and insulin receptor (INSR) are observed, suggesting that glycolysis is promoted [60]. In diabetic rats, CA administration causes weight loss and reduced biochemical parameters such as fasting blood glucose (FBG) levels, glucose tolerance, glycated serum proteins, total cholesterol, triglycerides, LDL, and FFA and increases HCL levels, emphasizing its hypoglycemic and hypolipidemic effects. Furthermore, a reduction in the level of MDA and ICAM-1 and an increase in SOD activity were observed, indicating that CA also has anti-inflammatory and anti-oxidant activities. Finally, the reduction of the insulin resistance index in diabetic rats treated with CA highlights its insulin-sensitizing effects [60].

Inflammation of adipose tissue in patients with diabetes and metabolic syndrome causes insulin resistance. As mentioned above, CA may improve insulin sensitivity by amplifying insulin signal transduction and stimulating the AMPK signaling pathway in adipose tissue. A study in mice fed a high-fat diet showed that CA can suppress IKKβ phosphorylation and reduce gene expression of pro-inflammatory cytokines, relieving inflammation of adipose tissue. It also enhances insulin signal transduction through modifications of Ser/Thr phosphorylation of IRS-1 and Akt [61]. A study in animal models of metabolic syndrome reports that CA treatment reduces blood pressure and serum free fatty acids (FFAs). Additionally, a reduction in markers of oxidative stress and inflammation, such as myeloperoxidase and C-reactive protein, was observed. This strengthens the role of CA in improving the inflammatory state typical of metabolic syndrome and preventing possible complications, such as atherosclerosis [45] (Figure 2).

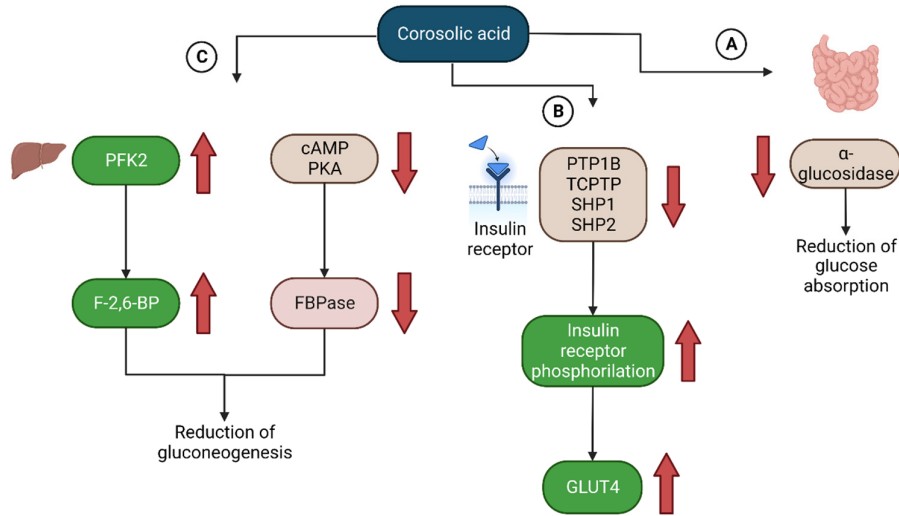

**Figure 2.** Anti-diabetic activity. (**A**). In the small intestine, CA binds to α-glucosidase and suppresses its catalytic function. (**B**). CA may inhibit the activity of certain PTPs such as protein tyrosine phosphatase 1B (PTP1B), T-cell protein tyrosine phosphatase (TCPTP), Src homology phosphatase-1 (SHP1), and Src homology phosphatase-2 (SHP2), thereby increasing the phosphorylation of the β subunit of the insulin receptor, and consequently increase the presence of GLUT4 receptors on the cell membrane. (**C**). CA increases F-2,6-BP production by decreasing cAMP levels and inhibiting PKA. This results in a lack of FBPase activation and an increase in PFK-2 activity, which then leads to an increase in F-2,6-BP synthesis. By acting in this way, CA causes an inhibitory effect on gluconeogenesis.

### 3.3. Anti-Tumor Activity

In recent years, several in vitro and in vivo experimental studies have demonstrated that CA seems to possess anti-tumor activity by acting on various tumorigenic processes, such as cell proliferation, apoptosis, angiogenesis, lymphangiogenesis, metastasis, and tumor immunity. These effects are achieved by modulation of several cancer-related signaling pathways, such as nuclear factor-kappa B (NF-κB), phosphatidylinositol 3 kinase/protein kinase B (PI3K/Akt) and Wnt/β-catenin pathways, nuclear factor erythroid 2, and many other components associated with cell proliferation or mortality. Furthermore, CA could exert a synergistic effect when administered with other anti-cancer agents [30].

In HER2-positive gastric cancer cells, dose- and time-dependent CA administration inhibits HER2 expression, arrests cell proliferation, and increases apoptotic cell death. Furthermore, the combination with adriamycin and 5-fluorouracil enhances the inhibitory effect on cell growth [62]. In prostate cancer TRAMP-C1 cells, CA epigenetically restores the expression of nuclear factor erythroid 2-related factor 2 (Nrf2), which results in protection against inflammatory stress [63]. In A549 cells, a human lung adenocarcinoma cell line, CA causes sub-G1 cell cycle arrest and caspase-dependent apoptotic cell death in a dose- and time-dependent manner by altering anti-apoptotic proteins through increased ROS levels [64].

### 3.4. Neuroprotective Properties

Zhang and colleagues investigated the neuroprotective effects of CA on cerebral ischemia–reperfusion (CIR) in rats. One week of oral administration of CA after surgery improved the damaged neuronal function and increased anti-oxidant activity with reduced ROS production and elimination of necrotic neurons. CA reduced levels of malondialdehyde (MDA) and 8-OHdG and has an anti-oxidant effect by increasing levels of glutathione peroxidase, superoxide dismutase (SOD), glutathione reductase, and CAT, providing neuroprotection against CIR injury. Furthermore, CA significantly reduced the levels of IL-1β, IL-6, TNF-α, PGE2, COX-2, and NO, while increasing the levels of IL-10 [65].

*3.5. Effects on Dyslipidemia and Hepatic Steatosis*

Takagi and colleagues evaluated the inhibitory effects of CA on liver steatosis and dietary hypercholesterolemia using animal models of type 2 diabetes mellitus (T2DM). Using two types of high-cholesterol diets with or without CA, the CA-treated group had a 32% reduction in mean blood cholesterol levels and a 46% reduction in liver cholesterol content after 10 weeks compared to the group that did not receive CA. This highlights the anti-dyslipidemic effect of CA [66]. Additionally, in a 4 h oral cholesterol load test, CA inhibited mean blood cholesterol levels compared to the control group. This suggests a direct effect of CA on the absorption of cholesterol in the small intestine by inhibiting the activity of cholesterol acyltransferase, which esterifies cholesterol in the small intestine. Failure to esterify cholesterol prevents it from being transported into the bloodstream, which explains the low cholesterol levels after CA administration [66]. Yamada and colleagues demonstrated that administering CA for 9 weeks to obese mice resulted in a 10% reduction in body weight and a 15% reduction in total fat mass, as well as reductions in glucose, insulin, and triglyceride levels [57]. CA treatment increased the expression of peroxisome proliferator-activated receptor (PPAR)-α in the liver and PPAR-γ in white adipose tissue (WAT), improving liver steatosis and insulin sensitivity by increasing plasma adiponectin levels and AdipoR1 in WAT. Upregulation of PPAR-α increases fatty acid β-oxidation and reduces triglyceride accumulation in the liver [67], while increased PPAR-γ expression prevents adipocyte hypertrophy [68].

CA can also ameliorate non-alcoholic steatohepatitis through its anti-inflammatory effects. In a study of mice fed a high-fat diet, CA reduced serum AST, ALT, triglycerides, and total cholesterol levels, and improved inflammatory indices and liver fibrosis. These effects are due to the inhibitory effects of CA on the release of pro-fibrogenic markers, including α-smooth muscle actin (α-SMA), collagen I, tissue inhibitor of metalloproteinase 1 (TIMP-1), and the pro-inflammatory cytokines transforming growth factor (TNF)-α, interleukin (IL)-1β, caspase-1, and IL-6. CA increases AMPK phosphorylation, which in turn inhibits lipid accumulation and attenuates the inflammatory response by blocking the NF-κB signaling pathway and thereby the release of inflammatory factors. Finally, CA inhibits mothers against decapentaplegic homolog 2 (Smad2) phosphorylation and TGF-β1 expression in liver tissue, resulting in a reduction of fibrosis [69]. Another study evaluated the hepatoprotective effects of ethanolic Banaba leaf extract (EBLE), containing approximately 12.87 mg CA, against dapsone-induced hepatotoxicity in rats. Dapsone administration increased serum hepatotoxicity marker enzymes, lipid peroxidation, and pro-inflammatory markers TNF-α and NF-κB and decreased anti-oxidants in liver tissue (superoxide dismutase, catalase, and glutathione). The administration of EBLE, together with silymarin, attenuates these abnormalities, in particular the expression of TNF-α and TGF-β, suggesting their possible use in cases of hepatotoxicity [70] (Table 2).

**Table 2.** Summary of the effects of corosolic acid (CA) from in vitro and animal studies.

| Authors and Year of Publication | Main Findings |
| --- | --- |
| **Hypercholesterolemia and Hepatic Steatosis** | |
| Takagi et al., 2010 [66] | CA reduces hypercholesterolemia and hepatic steatosis caused by dietary cholesterol in T2DM mice and may inhibit the activity of cholesterol acyltransferase in the small intestine |
| Yamada et al., 2008 [58] | CA reduces hepatic steatosis in obese mice by increasing PPAR-α expression in liver and PPAR-γ expression in WAT and ameliorates insulin sensitivity by increasing plasma adiponectin levels and AdipoR1 in WAT |
| Liu et al., 2021 [69] | CA reduces inflammation and fibrosis in NASH by regulating AMPK signaling pathways, NF-κB, and TGF-β1/Smad2 |

**Table 2.** *Cont.*

| Authors and Year of Publication | Main Findings |
|---|---|
| Singh and Ezhilarasan, 2022 [70] | EBLE containing CA and silymarin reduces hepatotoxicity through their anti-inflammatory and anti-oxidant effect |
| Lin et al., 2014 [42] | CA regulates IRAK1 phosphorylation via an NF-κB-independent pathway and plays a role in the inhibitory effect on acute inflammation |
| **Insulin resistance, metabolic syndrome, and prediabetes** | |
| Yang et al., 2016 [61] | CA suppresses phosphorylation of IKKβ and reduces gene expression of pro-inflammatory cytokines and enhances insulin signal transduction through modification of Ser/Thr phosphorylation of IRS-1 and Akt and stimulating the AMPK signaling pathway in adipose tissue |
| Yamaguchi et al., 2006 [45] | CA ameliorates hypertension, oxidative stress, and the inflammatory state in mice with metabolic syndrome |
| **Diabetes** | |
| Ni et al., 2019 [48] | CA could exert an inhibitory effect on α-glucosidase in a non-competitive and reversible manner through binding to the enzyme and causing a conformational change that interferes with its catalytic action |
| Zhang et al., 2017 [49] | CA in combination with acarbose inhibits α-amylase and α-glucosidase in a non-competitive manner |
| Xu et al., 2019 [60] | CA ameliorates hyperglycemia, hyperlipidemia, and insulin resistance in T2DM models through decreasing the expression of PEPCK and other genes involved in glucose metabolism, oxidative stress, and inflammation related to T2DM |
| **Cardioprotective effects in diabetes** | |
| Alkholifi et al., 2023 [46] | CA through the PPAR-γ pathway exerts cardioprotective and anti-oxidant effects on myocardial tissue in diabetic mice with acute myocardial infarction |
| **Kidney protection in diabetes** | |
| Li et al., 2016 [43] | CA inhibits the proliferation of diabetic glomerular mesangial cells via NADPH/ERK1/2 and p38 MAPK signaling pathways and prevents renal damage in diabetic animals |

Abbreviations. CA: corosolic acid; EBLE: ethanolic Banaba leaf extract; ERK: extracellular signal-regulated kinase; IRAK1: interleukin 1 receptor-associated kinase 1; IRR: inhibitor of nuclear factor-κB (IκB) kinase; IRS-1: insulin receptor substrate 1; MAPK, mitogen-activated protein kinase; NADPH: nicotinamide adenine dinucleotide phosphate; NASH, non-alcoholic steatohepatitis; PPAR, peroxisome proliferator-activated; Smad: mothers against decapentaplegic; T2DM: type 2 diabetes mellitus; TGF: transforming growth factor; WAT: white adipose tissue.

## 4. Effect of Corosolic Acid: Clinical Evidence

Leaves of *Lagerstroemia speciosa* L., containing CA, are used in folk medicine in several south-east Asian countries as a treatment for diabetes [40]. Therefore, in recent years, clinical studies have been conducted to understand the therapeutic properties of CA in humans.

Judy and colleagues examined the hypoglycemic effects of Banaba extracts standardized to 1% CA in patients with T2DM. Soft gel capsules containing 0.32 and 0.48 mg CA, as well as other compounds such as tannins, were administered once daily for 2 weeks. Posttreatment blood glucose levels were reduced by 30% in patients who received the 0.48 mg dose of CA. However, it is necessary to understand whether the hypoglycemic effects are due to CA alone or to the combination of this molecule with the other components of Banaba extract [71]. In another study, 12 patients with impaired FBG were treated with capsules containing 10 mg of CA from Banaba extract for 2 weeks. After treatment, fasting and 1 h postmeal glucose levels showed a 12% reduction. In addition, an average loss of 3 kg was observed after two weeks of treatment. However, despite the high levels of CA administered, even this study does not allow us to understand whether the observed effects are due to CA alone or to the combination with other components present in the extract [72]. The same product used in the latter study was used in an unpublished clinical

study by Xu ("Action of helping lower blood glucose level-clinical test", Chinese Center for Disease Control and Prevention, Beijing Hospital, 2008). One hundred patients with prediabetes or T2DM were enrolled and half of them were treated with CA 10 mg, while the other half received a placebo. After one month of treatment, CA-treated patients had a 10% reduction in fasting and 2 h postmeal blood sugar levels compared to the control group. Furthermore, a reduction in symptoms associated with diabetes such as hunger, thirst, and drowsiness has been reported. Moreover, no adverse events such as changes in blood pressure or liver and kidney function were reported [73].

In a double-blind study, the effects of 10 mg CA administered before a 75 g oral glucose tolerance test in patients with prediabetes or diabetes were evaluated. Blood glucose was measured every 30 min for 2 h. Patients who received CA showed a reduction in blood glucose levels at 60 and 120 min, with a statistically significant difference at 90 min compared to the control group. According to the authors, the compound used had 99% CA, so the decrease in blood glucose levels was specifically attributable to CA [74].

In a randomized trial of overweight and prediabetic patients, the effects of treatment with soybean leaf extracts (SLEs) and Banaba extracts, containing 0.3% CA, were evaluated compared to the placebo group after 12 weeks. Patients treated with SLE and Banaba extracts had a significant reduction in FBG levels. Furthermore, these patients had a significant reduction in glycosylated hemoglobin (HbA1c) levels compared to the group receiving a placebo, indicating that these compounds may influence long-term hyperglycemia and thus its complications. Patients who received SLE and Banaba extracts also had a reduction in HOMA-IR, a marker used to assess insulin resistance. This effect could be due to the anti-adiposity effects of these compounds, as excess adipose tissue is related to insulin resistance. In addition, a reduction in AST and ALT levels, markers of liver function, the increase in which indicates liver damage or disease, has also been observed. Therefore, it cannot be ruled out that the beneficial effects of SLE and Banaba extract on blood glucose levels and insulin resistance are due in part to the improvement of hepatocellular function in addition to the reduction of body fat. Finally, patients treated with Banaba extracts had a significant reduction in blood triglyceride levels compared to placebo, while the effects on total, HDL, and LDL cholesterol were not significantly different [75] (Table 3).

**Table 3.** Summary of the effects of corosolic acid (CA) in human studies.

| Authors and Year of Publication | Main Findings |
|---|---|
| Judy et al., 2003 [71] | Banaba extract containing 1% CA results in a 30% reduction in blood glucose levels after 2 weeks of treatment in patients with T2DM |
| Tsuchibe et al., 2006 [72] | 10 mg of CA from Banaba extract reduces fasting and postprandial 1 h blood glucose by 12 percent in addition to resulting in a reduction in body weight of about 3 kg after 2 weeks of treatment in patients with impaired fasting blood glucose |
| Xu et al., 2008 (unpublished) | 10 mg of CA reduced fasting and postprandial 2 h blood glucose levels by 10% after one month of treatment in patients with T2DM. In addition, a reduction in symptoms associated with diabetes was observed |
| Fukushima et al., 2006 [74] | 10 mg of CA before a 75 g oral glucose tolerance test reduces blood glucose levels at 60, 90, and 120 min in patients with prediabetes or diabetes |
| Choi et al., 2014 [75] | Banaba extract containing 0.3% CA reduces fasting blood glucose, HbA1c, HOMA-IR, AST, ALT, and blood triglyceride levels in overweight and prediabetes patients |

Abbreviations. CA: corosolic acid; HbA1c: glycated hemoglobin; HOMA-IR: homeostatic model assessment for insulin resistance; ALT: alanine transaminase; AST: aspartate aminotransferase; T2DM, type 2 diabetes mellitus.

## 5. Conclusions

CA appears to have anti-inflammatory, anti-dyslipidemic, and anti-diabetic properties, making it a potential therapeutic agent for patients with metabolic diseases. Its ability to improve lipid profiles, enhance glucose metabolism, increase insulin sensitivity, and reduce inflammation and oxidative stress highlights its importance in the management

of metabolic conditions. However, double-blind randomized controlled trials are needed to fully evaluate the efficacy and mechanisms of action and, thus, to establish CA as a standardized therapeutic option for these metabolic diseases.

**Author Contributions:** Conceptualization, R.C. and A.E.C.; methodology, V.G.; investigation, V.G.; data curation, V.G. and R.C.; writing—original draft preparation, V.G.; writing—review and editing, R.C. and A.E.C.; supervision, A.E.C.; project administration, R.C. and A.E.C.; All authors have read and agreed to the published version of the manuscript.

**Funding:** This research received no external funding.

**Conflicts of Interest:** The authors declare no conflict of interest.

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
