# Peer review of "Anti-Dyslipidemic and Anti-Diabetic Properties of Corosolic Acid: A Narrative Review"

_endocrines, doi:10.3390/endocrines4030044_

Round 1

Reviewer 1 Report

The authors of this well-written narrative review address an interesting topic regarding the potential therapeutic properties of Corosolic acid (CA), a nutraceutical derived from the Banaba tree (Lagerstroemia speciosa), in the management of metabolic diseases, such as hyperglycemia, overweight, and dyslipidemia. This manuscript carefully and extensively describes the mechanisms of action, the metabolism, and the potential therapeutical properties of CA. The use of nutraceuticals for the management of metabolic diseases has several limitations (paucity of scientific evidence, potential interactions with other drugs, lack of regulation and quality control measures), but these are well acknowledged in the manuscript. I just think that the strength of some sentences about the metabolic effects of CA should be attenuated.

Specific issues

1)      Since the evidence regarding the therapeutical effects of CA on diabetes in human is still weak, I would suggest a more cautious language in some parts of the manuscript (eg.: Line 211 “The [potential] antidiabetic activity of CA is [would be] exerted in several ways”; etc.). The same for its potential anti-cancer activities (“CA possesses [seems to possess] anti-tumor activity”, “CA exerts [could exert] a synergistic effect when administered with other anti-cancer agents”).

Minor issues

1)      Introduction (line 45): metabolic syndrome can be replaced by “MetS” (already used acronym)

2)      Introduction (lines 86-87): better replace “metabolic compensation” with “metabolic control”

3)      Introduction (lines 86-87): “…inhibit enzymes involved in carbohydrate metabolism, such as α-amylase and α-glucosidase”. Better replace with “carbohydrate absorption”

4)      Biological properties (lines 155-156): “making it a good candidate for the management of diabetes and related metabolic disorders”. Better replace with “good candidate as a help for the management…”

5)      Table 2: the column “main findings” shows some grammatical errors [CA reduce, CA improve, ameliorate… (reduces-improves-ameliorates?] and some typos (“CA reduce hypercholesterolemia and hepatic steatosis caused by dietary cholesterol in T2DM mice [and?] may inhibit the activity of cholesterol acyltransferase in the small intestine”)

6)      Lines 422-423: “Banaba leaves of (Lagerstroemia speciosa L.), containing CA, are used in folk medicine in several south-east Asian countries as a treatment for diabetes.” Please, add a ref.

7)      Along the manuscript, some typos should be corrected (eg., Line 39 or Lines 447-448).

8)      In several paragraphs of the manuscript concerning the biological properties of CA and inflammation [eg. Line 98, Line 157, Line 178, and Line 283], the Authors mention the ADA-EASD 2022 Consensus Report on diabetes management [ref. 15]. It looks as if the reference is wrong, since it has little to do with diabetes pathophysiology or with CA pharmacology, in my opinion. Please, check.

Author Response

Answers to the Reviewer #1 comments

Manuscript ID endocrines-2549099

Comment 1: The authors of this well-written narrative review address an interesting topic regarding the potential therapeutic properties of Corosolic acid (CA), a nutraceutical derived from the Banaba tree (Lagerstroemia speciosa), in the management of metabolic diseases, such as hyperglycemia, overweight, and dyslipidemia. This manuscript carefully and extensively describes the mechanisms of action, the metabolism, and the potential therapeutical properties of CA. The use of nutraceuticals for the management of metabolic diseases has several limitations (paucity of scientific evidence, potential interactions with other drugs, lack of regulation and quality control measures), but these are well acknowledged in the manuscript. I just think that the strength of some sentences about the metabolic effects of CA should be attenuated.

Answer to comment 1: Thank you so much. We appreciate your advice regarding the reshaping of some sentences.

Comment 2: Since the evidence regarding the therapeutical effects of CA on diabetes in human is still weak, I would suggest a more cautious language in some parts of the manuscript (eg.: Line 211 “The [potential] antidiabetic activity of CA is [would be] exerted in several ways”; etc.). The same for its potential anti-cancer activities (“CA possesses [seems to possess] anti-tumor activity”, “CA exerts [could exert] a synergistic effect when administered with other anti-cancer agents”).

Answer to comment 2: We have followed your advice and the sentences you suggested on the effects of corosolic acid have been reshaped.

Comment 3: Introduction (line 45): metabolic syndrome can be replaced by “MetS” (already used acronym).

Answer to comment 3: Corrected as requested.

Comment 4: Introduction (lines 86-87): better replace “metabolic compensation” with “metabolic control”.

Answer to comment 4: Thanks for the advice. We changed it.

Comment 5: Introduction (lines 86-87): “…inhibit enzymes involved in carbohydrate metabolism, such as α-amylase and α-glucosidase”. Better replace with “carbohydrate absorption”.

Answer to comment 5: Thanks for the advice. We changed it.

Comment 6: Biological properties (lines 155-156): “making it a good candidate for the management of diabetes and related metabolic disorders”. Better replace with “good candidate as a help for the management…”.

Answer to comment 6: Thanks for the advice. We changed it.

Comment 7: Table 2: the column “main findings” shows some grammatical errors [CA reduce, CA improve, ameliorate… (reduces-improves-ameliorates?] and some typos (“CA reduce hypercholesterolemia and hepatic steatosis caused by dietary cholesterol in T2DM mice [and?] may inhibit the activity of cholesterol acyltransferase in the small intestine”).

Answer to comment 7: The grammatical errors in Table 2 have been corrected.

Comment 8: Lines 422-423: “Banaba leaves of (Lagerstroemia speciosa L.), containing CA, are used in folk medicine in several south-east Asian countries as a treatment for diabetes.” Please, add a ref.

Answer to comment 8: The reference has been added. It is the #22 of the revised version of the manuscript.

Comment 9: Along the manuscript, some typos should be corrected (eg., Line 39 or Lines 447-448).

Answer to comment 9: Thank you for spotting these typos which have been corrected in the revised version of the manuscript.

Comment 10: In several paragraphs of the manuscript concerning the biological properties of CA and inflammation [eg. Line 98, Line 157, Line 178, and Line 283], the Authors mention the ADA-EASD 2022 Consensus Report on diabetes management [ref. 15]. It looks as if the reference is wrong, since it has little to do with diabetes pathophysiology or with CA pharmacology, in my opinion. Please, check.

Answer to comment 10: Sorry for the confusion. There was an error uploading the references in the final version of the manuscript, which has now been corrected.

Reviewer 2 Report

Thank you for giving me the opportunity to review this article. The Authors reviewed Corosolic acid (CA) from different aspects, including pharmacodynamics, preclinical and clinical studies. The information in this review is very insightful but requires some corrections and reorganization.

Overall comment:

1-The authors have introduced a new definition not commonly used in the scientific literature or clinical practice "Dysmetabolic disorders. "

Dysmatbolic disorder might lead the reader to think of dysmetabolic syndrome ( which is an uncommon name for metabolic syndrome or syndrome X )

Please change "Dysmetabolic disorder" throughout the document. Alternatively, introduce it as a new terminology and add the reference of your source.

2-References and citations throughout the document do not match. There are 57 citations and 69 references.

Furthermore, the text does not match the cited reference.

Example ( please note this is not the entire list. All references need to be checked)

a-citation [1] should be WHO, but it is 2 in the ref list

b-citation [13] and [14]  should be about CA, but they are guidelines in the ref list

c- ref 12, 20 about diabetes but cited with the biological effect of CA

d-Xu is cited as 47, but it is 60 in the ref list

e-Judy is cited as 49, but it is 61 in the ref list

So kindly revise all citations and references. "

3- Many double Spaces between sentences need to be corrected.

Section comment

Abstract:

Please remove the section regarding PCO, as the authors did not add any insight regarding this disease.

Introduction

a-as mentioned above: Dysmetabolic disorders either change or add a reference to your source

b-Line 54 Add reference.

c- Move Table 1 after line 64 (Main nutraceutical compounds available and approved for clinical practice are  reported in Table 1.)

d-Table 1 Add a new column and add references( with proper journal citations, not only the names)  to each row as this information is new and not cited anywhere else.

e- Add ref line 84 "It has attracted considerable attention for its potential health benefits and therapeutic properties, such as anti-diabetic, anti-obesity, anti-hyperlipidemic, anti-inflammatory, and anti-tumor effects.

Molecular mechanisms

1-     Lines 117-118 need rephrasing. It is challenging to ascertain the sentence's meaning.

2-     Expand more about absorption. How did they distinguish between stomach and intestinal absorption?

Effect of corosolic acid: Preclinical evidence

Start this section with Anti-inflammatory and antioxidant effects (2.3). Why were they added under molecular mechanisms?

-Figure 1: Modify the colors. Anything that is inhibited or downregulated should be in red, and activated or upregulated should be in green, e.g., figure 1 A IRKA1 is inhibited, so it should be in red

   - please modify the figures accordingly

-Lines 215-216 include contradicting information "Ni et al. have shown that CA is able to inhibit ?-glucosidase activity in a non-competitive manner. In fact, CA binds the active site of the enzyme."

Non-competitive means it binds to an allosteric site away from the active site, so how come it is non-competitive and binds to the active site ?? please correct

-Lines 221-222 authors keep referring to glucose reabsorption. Do they mean from the kidney?

Absorption: refer to material entering the body or bloodstream from the GI tract

Reabsorption renting the body from the kidney after filtration

If the author means on a cellular level, it should be referred to as uptake or reuptake

-Line 283 add another ref as this ref is not cancer-specific, and please add more details and some examples of cancers in the "Anti-tumor activity" section

-Line 306, as mentioned above, move and start it with antioxidants

- The paragraph starting 346-352 should be added to the anti-inflammatory section.

-line 404-411 repeated information

-Table 2 and Table 3 add the citations for your reference

-Different sections talk about insulin resistance and diabetes should be all one section, not scattered

-Add a section for lipid profile and move all information related to it there

Corosolic acid and polycystic ovary syndrome

-REMOVE PCO section not related to the paper author described the disease and related insulin resistance mechanism, followed by a repeated section on how CA affects insulin resistance.

I don't know if the intended citation has actual information about CA and PCO, but at the moment, the entire section of PCO seems out of place and unrelated to the entire review.

So please remove this section or offer an actual reference linking CA to PCO, not the author's assumption or point of view.

English is acceptable. 

Double spaces between sentences and punctuation need correction 

and they use some terminology inappropriate in some places 

Author Response

Answers to the Reviewer #2 comments

Manuscript ID endocrines-2549099

Comment 1: Thank you for giving me the opportunity to review this article. The Authors reviewed Corosolic acid (CA) from different aspects, including pharmacodynamics, preclinical and clinical studies. The information in this review is very insightful but requires some corrections and reorganization.

Answer to comment 1: Thank you. We appreciate the constructive criticisms of the Reviewer which allowed us to improve the quality of the manuscript.

Comment 2: The authors have introduced a new definition not commonly used in the scientific literature or clinical practice "Dysmetabolic disorders. "Dysmatbolic disorder might lead the reader to think of dysmetabolic syndrome (which is an uncommon name for metabolic syndrome or syndrome X). Please change "Dysmetabolic disorder" throughout the document. Alternatively, introduce it as a new terminology and add the reference of your source.

Answer to comment 2: Thank you. We have changed the wording “Dysmetabolic disorders” with “Metabolic disorders”.

Comment 3: References and citations throughout the document do not match. There are 57 citations and 69 references. Furthermore, the text does not match the cited reference. Example (please note this is not the entire list. All references need to be checked). a-citation [1] should be WHO, but it is 2 in the ref list. b-citation [13] and [14]  should be about CA, but they are guidelines in the ref list. c- ref 12, 20 about diabetes but cited with the biological effect of CA. d-Xu is cited as 47, but it is 60 in the ref list. e-Judy is cited as 49, but it is 61 in the ref list. So kindly revise all citations and references. "

Answer to comment 3:  Sorry for the confusion. There was an error uploading the references in the final version of the manuscript, which has now been corrected.

Comment 4: Many double Spaces between sentences need to be corrected.

Answer to comment 4: The few double spaces between sentences that we found were corrected.

Comment 5: Abstract: Please remove the section regarding PCO, as the authors did not add any insight regarding this disease.

Answer to comment 5: As suggested, the PCOS section has been removed from both the Abstract and the text.

Comment 6: Introduction: a-as mentioned above: Dysmetabolic disorders either change or add a reference to your source. b-Line 54 Add reference. c- Move Table 1 after line 64 (Main nutraceutical compounds available and approved for clinical practice are reported in Table 1.) d-Table 1 Add a new column and add references (with proper journal citations, not only the names) to each row as this information is new and not cited anywhere else. e- Add ref line 84 "It has attracted considerable attention for its potential health benefits and therapeutic properties, such as anti-diabetic, anti-obesity, anti-hyperlipidemic, anti-inflammatory, and anti-tumor effects.

Answer to comment 6: The phrase “Dysmetabolic disorders” has been changed to “Metabolic disorders”. The reference was added to line 52 [9]. Table 1 was moved after line 63 and the column containing the references has been added. The reference was added to line 86 [14].

Comment 7: Molecular mechanisms: 1-     Lines 117-118 need rephrasing. It is challenging to ascertain the sentence's meaning. 2-     Expand more about absorption. How did they distinguish between stomach and intestinal absorption?

Answer to comment 7: The sentence in lines 117-118 has been reworded to make it more understandable. Regarding absorption, study methods have been reported to distinguish between stomach and intestinal absorption.

Comment 8: Effect of corosolic acid: Preclinical evidence: Start this section with Anti-inflammatory and antioxidant effects (2.3). Why were they added under molecular mechanisms? -Figure 1: Modify the colors. Anything that is inhibited or downregulated should be in red, and activated or upregulated should be in green, e.g., figure 1 A IRKA1 is inhibited, so it should be in red.  - please modify the figures accordingly. -Lines 215-216 include contradicting information "Ni et al. have shown that CA is able to inhibit ?-glucosidase activity in a non-competitive manner. In fact, CA binds the active site of the enzyme." Non-competitive means it binds to an allosteric site away from the active site, so how come it is non-competitive and binds to the active site ?? please correct. -Lines 221-222 authors keep referring to glucose reabsorption. Do they mean from the kidney?

Absorption: refer to material entering the body or bloodstream from the GI tract. Reabsorption renting the body from the kidney after filtration. If the author means on a cellular level, it should be referred to as uptake or reuptake. -Line 283 add another ref as this ref is not cancer-specific, and please add more details and some examples of cancers in the "Anti-tumor activity" section. -Line 306, as mentioned above, move and start it with antioxidants. - The paragraph starting 346-352 should be added to the anti-inflammatory section. -line 404-411 repeated information -Table 2 and Table 3 add the citations for your reference. -Different sections talk about insulin resistance and diabetes should be all one section, not scattered .- Add a section for lipid profile and move all information related to it there.

Answer to comment 8: Figures have been modified accordingly. The section “Effect of corosolic acid: Preclinical evidence” was reworded as recommended and the section Effects on dyslipidemia and hepatic steatosis was added. The reported errors have been corrected and Figure 1 has been remodeled. Finally, in the paragraph “Anti-tumor activity” some examples concerning CA and its anti-tumor activity have been added.

Comment 9: Corosolic acid and polycystic ovary syndrome: -REMOVE PCO section not related to the paper author described the disease and related insulin resistance mechanism, followed by a repeated section on how CA affects insulin resistance. I don't know if the intended citation has actual information about CA and PCO, but at the moment, the entire section of PCO seems out of place and unrelated to the entire review. So please remove this section or offer an actual reference linking CA to PCO, not the author's assumption or point of view.

Answer to comment 9: Thanks for the advice. Reviewing the article actually the part about PCOS seems to be redundant and there is currently no evidence in the literature regarding the effects of corosolic acid on this condition. As advised, this section has been removed.

Round 2

Reviewer 2 Report

I want to thank the authors for their effort to address all the comments.

Minor modifications are still needed.

1-     Table 1: put the reference same style as Tables 2 and 3 with citations, e.g., Takagi et., al 2010 (49)

2-     Lines 236-238 still have the same problem. You cannot add two contradictory sentences together. Non-competitive means bind away from the enzyme's active site, so please review the original article you cited to confirm the author's actual meaning. If the original contains the same contradictory information, please remove it from your review.

You can remove one of the sentences, but you can't put them together

3-     Line 344, please add a full stop(.)

4-     Paragraphs 341-350 put all in one paragraph

Language needs some minor editions and proofreading.

Author Response

Answers to the Reviewer #2 comments

Manuscript ID endocrines-2549099

Comment 1. I want to thank the authors for their effort to address all the comments. Minor modifications are still needed.

Answer to comment 1. We thank the Reviewer for the words of appreciation for our study. We have carefully reviewed our manuscript based on the comments raised. The responses are listed one-to-one below.

Comment 2. Table 1: put the reference same style as Tables 2 and 3 with citations, e.g., Takagi et., al 2010 (49)

Answer to comment 2. Done as requested. Please see the revised Table 1.

Comment 3. Lines 236-238 still have the same problem. You cannot add two contradictory sentences together. Non-competitive means bind away from the enzyme's active site, so please review the original article you cited to confirm the author's actual meaning. If the original contains the same contradictory information, please remove it from your review. You can remove one of the sentences, but you can't put them together.

Answer to comment 3. We deleted the sentence (please see lines 241-242).

Comment 4. Line 344, please add a full stop(.)

Answer to comment 4. Done (please see line 342).

Comment 5. Paragraphs 341-350 put all in one paragraph

Answer to comment 5. Done (please see lines 339-346).

Comment 6. Comments on the Quality of English Language: Language needs some minor editions and proofreading.

Answer to comment 6. The manuscript has been carefully reviewed and typos have been corrected. Thank you.